# Sixth-Week Immune-Nutritional-Inflammatory Biomarkers: Can They Predict Clinical Outcomes in Patients with Advanced Non-Small Cell Lung Cancer Treated with Immune Checkpoint Inhibitors?

**Polat Olgun [1,2,*]** and **Omer Diker [1,2]**

1 Department of Medical Oncology, Faculty of Medicine, Near East University, 99138 Nicosia, Cyprus; omeromrumdiker@gmail.com

2 Medical Oncology, Dr. Burhan Nalbantoğlu Government Hospital, 99010 Nicosia, Cyprus

* Correspondence: olgunpolat100@gmail.com; Tel.: +90-(548)-860-9190

**Abstract:** Background: We investigated the relationships between inflammatory markers such as the neutrophil-to-lymphocyte ratio (NLR), platelet-to-lymphocyte ratio (PLR), Lung Immune Prognostic Index (LIPI), and modified Glasgow prognostic score (mGPS) to determine whether they could predict treatment response to pembrolizumab or nivolumab (immunotherapy) 6 weeks after the start of treatment (post-treatment). Methods: We included all patients with lung cancer treated with immunotherapy. We examined the biomarker trends and explored their associations with progression-free survival (PFS), overall survival (OS), and response rate (RR) at 6 weeks. Results: Eighty-three patients were enrolled in the study. The presence of liver metastasis, low post-treatment NLR (<5), low post-treatment PLR (<170), intermediate post-treatment LIPI, and immune-related adverse events were significantly associated with the response. The multivariate analysis revealed that high post-treatment NLRs $\geq$ 5 ($p = 0.004$) and PLRs $\geq$ 170 ($p \leq 0.001$) were independent prognostic factors of shorter OS. A good LIPI status was associated with better PFS ($p = 0.020$) and OS ($p = 0.065$). Post-treatment mGPS (0–2) was significantly associated with improved PFS ($p = 0.009$) and OS ($p = 0.064$). Conclusions: Post-treatment NLR, PLR, LIPI, and mGPS are associated with worse OS and recurrence. These findings should be independently and prospectively validated in further studies.

**Keywords:** immune-based prognostic scores; neutrophil-to-lymphocyte ratio (NLR); platelet-to-lymphocyte ratio (PLR); Lung Immune Prognostic Index (LIPI); modified Glasgow prognostic score (mGPS)



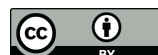

## 1. Introduction

Lung cancer is the most prevalent life-threatening malignancy and cause of cancer-related death worldwide [1]. The primary traditional treatments for patients with lung cancer are surgery, radiotherapy, and chemotherapy. Approximately 75% of cases are diagnosed at advanced stages, and the benefits achieved from chemotherapy and the prognosis of non-small cell lung cancer (NSCLC) remain poor in advanced stages. NSCLC is classified into three main histological types: adenocarcinoma, squamous cell carcinoma, and large cell carcinoma [2].

In recent years, immunotherapy has developed as a novel strategy for the management of NSCLC. Many studies have indicated that tumour cells can evade the anti-tumour responses of T cells and that programmed cell death (PD)-1 immune checkpoint inhibitors (ICIs) such as nivolumab and pembrolizumab inhibit programmed cell death protein-1 (PD-1)-mediated signalling by blocking its ligands (PD-L1 and PD-L2) [3]. Since 2015, anti-PD1 immunotherapy has been used as a gold standard treatment for stage IV NSCLC (as first- or second-line treatment) in combination with chemotherapy or monotherapy.

Oncogenic mutation drivers such as epidermal growth factor receptor (EGFR), anaplastic lymphoma kinase (ALK), and repressor of silencing 1 (ROS-1) can alter the immune tumour microenvironment, which may induce anti-PD1/PD-L1 resistance. However, anti-PD1/PD-L1 immunotherapy is highly effective for NSCLC with KRAS or BRAF mutations [4].

ICIs have been identified and clinically validated as predictive biomarkers such as PDL-1 expression, tumour mutational burden (TMB), and high microsatellite instability (MSI-H). In general, as the PD-L1 score increases, the treatment response rate (RR) also increases in lung cancer. On the other hand, PD-L1-positive tumours may not respond to treatment, and PD-L1-negative tumours can achieve a response in some patients. The tumour mutation burden (TMB), also called the total number of tumour mutations per megabase coding tumour genome, has been defined as a predictive marker of lung cancer. Many factors are considered in calculating the TMB, such as methodology, panel range, genome selection, mutation diversity, sequencing technologies, and bioinformatic algorithms [5]. Despite all these details, TMB cut-off values have not been clarified, and their clinical usefulness has not yet been recognised as a clear biomarker. Defects in DNA mismatch repair (dMMR) are defined as immunohistochemical protein loss. Next-generation sequencing (NGS) technology identifies tumours and mutations using a multiple analysis called microsatellite instability (MSI). dMMR or MSI-H detection is rare in patients with NSCLC [6]. However, these biomarkers have some limitations; therefore, physicians need an effective biomarker for risk stratification [7].

The tumour-associated inflammation in patients with cancer is believed to influence the host immune response and resistance, growth, and migration of tumours via certain inflammatory factors [8]. Due to the interplay between systemic inflammation, the immune system, and immunotherapy, the pre-treatment neutrophil-to-lymphocyte ratio (NLR), platelet-to-lymphocyte ratio (PLR), the Lung Immune Prognostic Index (LIPI), and modified Glasgow prognostic score (mGPS) have been indicated to predict the therapeutic effect or outcomes related to poor survival in patients with various cancers (gastrointestinal carcinomas, renal cancer, breast cancer, ovarian cancer, prostate cancer, etc.) [9–11].

In recent years, scientists have found that pre- and post-treatment changes in the composition of peripheral blood cells could reflect the body's anti-tumour status more accurately, thus affecting prognosis [12]. Little information is available (in 2019, Kasahara et al. [13]; in 2021, Küçükarda et al. [14]) regarding the association of the post-treatment inflammation parameter with response to anti-PD1 treatment in patients with advanced NSCLC.

In this study, we aimed to determine the prognostic usefulness of post-treatment NLR, PLR, LIPI, and mGPS obtained at the 6th week of treatment to reveal a reliable, robust, inexpensive, and potentially tumour-agnostic post-treatment indicator for predicting the response to anti-PD-1 combined therapy.

## 2. Materials and Methods

### 2.1. Study Population

We conducted a retrospective analysis of consecutive patients with NSCLC who had undergone anti-PD-1 antibody (nivolumab or pembrolizumab) and anti-CTLA4 (ipilimumab) treatments at Burhan Nalbantoglu Research Hospital and Near East University Hospital between March 2017 and March 2021. These hospitals are the two largest reference centres in North Cyprus.

### 2.2. Data Collection

The study inclusion criteria were as follows: (a) pathologically confirmed NSCLC; (b) initial-stage IIIB or IV, recurrence after curative surgery or maintenance after chemoradiotherapy; and (c) administration of nivolumab at 3 mg/kg every 2 weeks, pembrolizumab at a 200 mg flat dose every 3 weeks, or 4 cycles of nivolumab at 3 mg/kg and ipilimumab at 1 mg/kg every 3 weeks, followed by maintenance nivolumab at 3 mg/kg every 2 weeks as palliative therapy. All the patients received anti-PD-1 therapy, while some received additional anti-neoplastic therapies. Baseline patient and tumour characteristics are shown in Table 1.

**Table 1.** Baseline patient and tumour characteristics.

| Characteristic | |
|---|---|
| **Age at start of treatment (years)** | |
| Median | 66 |
| Range | 42–88 |
| <70 | 64 (69.5) |
| >70 | 28 (30.4) |
| **Sex** | *n* (%) |
| Male | 73 (88) |
| Female | 10 (12) |
| **ECOG performance status score** | *n* (%) |
| 0–1 | 46 (50.0) |
| 2–4 | 46 (50.0) |
| **Tumour histology** | *n* (%) |
| Squamous cell carcinoma | 32 (38.6) |
| Adenocarcinoma | 48 (57.8) |
| NSCL, NOS | 2 (2.4) |
| Adenosquamos cell carcinoma | 1 (1.2) |
| **PD-L1** | *n* (%) |
| Negative | 11 (13.3) |
| 1–49% | 3 (3.6) |
| ≥50% | 5 (6.0) |
| Unknown | 64 (77.1) |
| **Smoking status** | *n* (%) |
| Current or former smoker | 79 (95.2) |
| Never smoked | 4 (4.8) |
| **Immuno-therapy line** | *n* (%) |
| 1st | 41 (49.4) |
| 2nd | 41 (49.4) |
| 3rd | 1 (1.2) |
| Maintenance | |
| **Post-treatment neutrophil-to-lymphocyte ratio** | *n* (%) |
| Median | 14.15 (1.07–115.0) |
| <5 | 1 (1.9) |
| >5 | 91 (98.1) |
| **Post-treatment platelet-to-lymphocyte ratio** | *n* (%) |
| Median | 797.00 (4.00–3223.0) |
| <150 | 11 (10.9) |
| >150 | 82 (89.1) |
| **Post-treatment LIPI** | *n* (%) |
| GOOD | 26 (31.3) |
| INTERMEDIATE | 33 (39.8) |
| POOR | 23 (27.7) |
| Unknown | 1 (1.2) |
| **Post-treatment mGPS** | *n* (%) |
| Median | |
| 0 | 5 (6.0) |
| 1 | 29 (34.9) |
| 2 | 16 (19.3) |
| Unknown | 33 (39.8) |
| **Post-treatment immunotherapy-related adverse event** | *n* (%) |
| Non-adverse event | 66 (79.5%) |
| Adverse event | 17 (20.5%) |

**Table 1.** *Cont.*

| Characteristic | |
|---|---|
| Pre-treatment liver metastasis | *n* (%) |
| Median | |
| Non-liver | 63 (75.9) |
| Liver | 20 (24.1) |
| Immunotherapy type | *n* (%) |
| Pembrolizumab | 44 (53) |
| Nivolumab | 38 (45.8) |
| Nivolumab + ipilumab then nivolumab | 1 (1.2) |

N: Number; ECOG: Eastern Cooperative Oncology Group; NSCL: non-small cell lung cancer; NOS: not otherwise specified; Pdl1: platelet-derived lymphocyte; LIPI: Lung Immune Prognostic Index; mGPS: modified Glasgow prognostic score.

The exclusion criteria were as follows: patients who had received induction chemotherapy or immunotherapy, oncogene-addicted malignancies (EGFR mutations, ALK mutations, ROS mutations, etc.), and auto-immune or interstitial lung diseases requiring steroid therapy.

*2.3. Outcomes*

Complete blood cell counts and biochemistry parameters were assessed before each drug administration. Total white blood cell (WBC) counts, absolute neutrophil counts (ANC), absolute lymphocyte counts (ALC), platelet (PLT) counts, and levels of lactate dehydrogenase (LDH), albumin, and C-reactive protein (CRP) were analysed 6 weeks after the start of treatment.

The NLR was calculated as the ratio of ANC to ALC, and an NLR $\geq 5$ was considered high [15]. PLR was defined as the ratio of PLT to ALC and categorised using a threshold value of $\geq 150$ [16].

LIPI scores were grouped according to the derived NLR (dNLR) and LDH levels. The dNLR was calculated as the ratio of ANC to [WBC count $-$ ANC]. The LIPI scores were stratified into three risk groups: good (dNLR < 3 + LDH < upper limit of normal [ULN]), intermediate (dNLR > 3 or LDH > ULN), and poor risk (dNLR > 3 + LDH > ULN) [17].

An mGPSs of 0, 1, or 2 was given depending on the CRP (>1.0 mg/dL) and albumin levels (<35 g/L): mGPS0 = albumin (>35 g/L) and CRP (<1.0 mg/dL), or albumin (<35 g/L) and CRP (<1.0 mg/dL); mGPS1 = albumin (>35 g/L) and CRP (>1.0 mg/dL); and mGPS2 = albumin (<35 g/L) and CRP (>1.0 mg/dL) [18].

Tumour PD-L1 expression was detected using immunoassay with the Ventana PD-L1 monoclonal antibody SP263 (Roche, Basel, Switzerland).

Thoracoabdominopelvic computed tomography (CT) or positron emission tomography (PET)-CT scans were performed every 12 weeks in accordance with the study protocol, additionally as needed depending on patient clinical status to assess earlier disease progression. All responses were evaluated on the basis of the revised Response Evaluation Criteria in Solid Tumours (RECIST) guideline (version 1.1).

Hyperprogression, as determined using the RECIST guideline, is an unexpected rapid disease progression under immunotherapy compared with baseline in the first evaluation after immunotherapy with a $\geq 2$-fold increase in growth rate. Pseudoprogression is the phenomenon in which an initial increase in target lesion length occurs or new lesions appear, followed by tumour shrinkage; these changes can be observed using a tumour biopsy or continuous radiography scan [19].

*2.4. Statistical Analysis*

All analyses were conducted using SPSS (version 22.0). The Cox proportional hazards model was used to test the relationships between the variables and PFS, OS, and RR. *p* values < 0.05 were considered statistically significant in all the analyses. PFS describes the time from the start of immunotherapy treatment to the date of disease progression or death.

OS is the period from the date of starting immunotherapy treatment that resulted in patient survival or death. ORR describes the percentage of responses among all the treated patients. In univariate analysis, variables who had *p* value < 0.05 were include in the multivariate model.

## 3. Results

### 3.1. Patient Demographics

This study included 83 patients with advanced NSCLC (88% men and 12% women); their demographic characteristics are shown in Table 1. The median age at treatment onset was 66 years (range: 42–88 years). Of the patients, 95.2% were smokers, and 50% had an Eastern Cooperative Oncology Group performance status of $\geq 2$. The present study group included 57.8% patients with adenocarcinoma, 38.6% with squamous cell carcinoma, two with poorly differentiated carcinoma, and one with adenosquamous carcinoma. One patient had EGFR mutations, one showed ALK rearrangements, and none had c-ROS oncogene (ROS 1) rearrangement or BRAF gene mutations. According to the metastasis site, 24.1% of the patients had liver metastasis. All the patients received immunotherapy, 53% received pembrolizumab, 45.8% received nivolumab, and one received nivolumab plus ipilimumab therapy. According to the treatment sequence, 41 patients ranked it first, and 41 ranked it second. As the first-line treatment, the patients with pDL1 $\geq 50\%$ received mono immunotherapy, while the patients with pDL1 < 50% received maintenance immunotherapy after four cycles of chemotherapy and immunotherapy. The results showed that 20.5% of the patients had immune-related adverse effects.

### 3.2. Immunologic Biomarkers

The biomarker results after the third nivolumab or second pembrolizumab infusions are presented in Table 1. The sixth-week blood counts, NLR, PLR, LIPI, and mPGS were investigated.

The patients were divided into two groups with the threshold value of NLR < 5 at 6 weeks post-treatment, which was associated with poorer PFS. According to the NLR reduction, the anti-PD-1 antibody treatment was associated with a higher objective RR (HR = 0.703; 95% confidence interval (CI), 0.556–0.888; *p* = 0.003) and a significantly improved PFS (HR = 1.162; 95% CI, 1.091–1.237; *p* < 0.001) and OS (HR = 1.182; 95% CI: 1.107–1.261; *p* < 0.001). All results were statistically significant (Tables 2 and 3). We performed multivariate analyses to determine the prognostic importance of the clinical characteristics of NLR for improved PFS (HR = 1.212; 95% CI, 0.924–1.59; *p* = 0.165), which was not statistically significant, but the improved OS was statistically significant (HR = 1.456; 95% CI, 1.128–1.880; *p* = 0.004).

This study also reported the relationship between PLR levels and RR, PFS, and OS in patients with cancer treated with immunotherapy. PLR thresholds (<170 and $\geq 170$) were used as the factors of the sub-group analysis. The univariate analysis for PFS (HR = 1.001; 95% CI, 1.001–1.002; *p* = 0.001) and OS (HR = 0.996; 95% CI, 0.994–0.999; *p* = 0.001) showed an additional significant association with PLR. Multivariate analysis for higher PLR at baseline was associated with shorter PFS (HR = 0.998; 95% CI, 0.995–1.001; *p* = 0.188) and an independent prognostic factor of OS (HR = 0.996; 95% CI, 0.994–0.999; *p* = 0.001).

We then investigated the association between different LIPI cut-off values as the patient's clinical outcomes and the prognostic value of post-treatment LIPI for OS. The cut-off values were as follows: intermediate OS group (HR = 1.650; 95% CI, 0.913–2.982; *p* = 0.097) and poor OS group (HR = 2.086; 95% CI, 1.111–3.916; *p* = 0.022); intermediate PFS group (HR = 1.985; 95% CI, 1.115–3.534; *p* = 0.020) and poor PFS group (HR = 2.24, 95% CI, 1.214–4.156; *p* = 0.010); intermediate RR group (HR = 0.138; 95% CI, 0.24–1.82; *p* = 0.001); and the poor RR group (HR = 0.196; 95% CI, 0.17–2.10; *p* = 0.009). The univariate analysis revealed that elevated LIPI was statistically significantly related to an independent prognostic factor of OS, PFS, and RR, except in the intermediate group. The multivariate analyses revealed that only intermediate LIPI (HR = 0.188, 95% CI, 0.053–0.664; *p* = 0.009) was a significant prognostic factor (Tables 2 and 3).

**Table 2.** Univariable and multivariable analyses of overall survival and progression-free survival.

| Variables | Progression-Free Survival | | Overall Survival | |
|---|---|---|---|---|
| | Unadjusted HR (95% CI), *p* | Adjusted HR (95% CI), *p* | Unadjusted HR (95% CI), *p* | Adjusted HR (95% CI), *p* |
| ECOG PS ≥ 2 | 1.48 (0.925–2.383) 0.101 | - | 1.61 (0.989–2.640) 0.056 | - |
| Histology-Non-SQ | 0.58 (0.343–1.000) 0.050 | - | 0.69 (0.402–1.187) 0.180 | - |
| >75 years of age | 0.98 (0.96–1.006) 0.137 | | 0.55 (0.969–1.017) 0.993 | |
| Presence of brain metastasis | 1.16 (0.502–2.688) 0.725 | - | 1.144 (0.458–2.855) 0.773 | - |
| Presence of bone metastasis | 1.383 (0.844–2.26) 0.199 | | 1.18 (0.718–1.94) 0.512 | |
| Presence of adrenal gland metastasis | 1.51 (0.849–2.685) 0.161 | - | 1.592 (0.876–2.895) 0.127 | - |
| Presence of malignant pleural metastasis | 1.931 (1.062–3.508) 0.031 | 1.675 (0.563–4.982) 0.354 | 1.223 (0.666–2.248) 0.516 | - |
| Presence of liver metastasis | 1.994 (1.179–3.373) 0.010 | 3.093 (1.017–9.405) 0.047 | 2.060 (1.195–3.550) 0.009 | 1.97 (0.926–4.21) 0.078 |
| irAEs | 0.444 (0.241–0.817) 0.009 | 0.339 (0.086–1.339) 0.123 | 0.523 (0.279–0.981) 0.043 | 0.120 (0.036–0.402) 0.001 |
| NLR | 1.162 (1.091–1.237) 0.000 | 1.212 (0.924–1.59) 0.165 | 1.182 (1.107–1.261) 0.000 | 1.456 (1.128–1.880) 0.004 |
| PLR | 1.001 (1.001–1.002) 0.001 | 0.998 (0.995–1.001) 0.188 | 1.001 (1.000–1.002) 0.003 | 0.996 (0.994–0.999) 0.001 |
| LIPI status | N/A 0.002 | N/A 0.940 | N/A 0.065 | N/A 0.786 |
| • Good | 1 | 1 | 1 | 1 |
| • Intermediate | 1.985 (1.115–3.534) 0.020 | 0.802 (0.213–3.01) 0.744 | 1.650 (0.913–2.982) 0.097 | 0.731 (0.246–2.17) 0.928 |
| • Poor | 2.24 (1.214–4.156) 0.010 | 0.729 (0.111–4.80) 0.743 | 2.086 (1.111–3.916) 0.022 | 0.837 (0.245–2.86) 0.576 |
| Pdl1 | N/A 0.395 | | N/A 0.779 | |
| • 0 | 1 | | 1 | N/A |
| • 1–49 | 0.342 (0.72–1.616) 0.176 | | 0.848 (0.226–3.181) 0.807 | |
| • >50 | 0.753 (0.250–2.270) 0.614 | | 1.381 (0.454–4.199) 0.570 | |
| CRP level | 1.097 (1.034–1.165) 0.002 | 1.015 (0.916–1.125) 0.771 | 1.032 (0.978–1.089) 0.246 | |
| mGPS | N/A 0.009 | N/A 0.373 | N/A 0.064 | N/A 0.180 |
| • 0 | 1 (reference) | 1 (reference) | 1 (reference) | 1 (reference) |
| • 1 | 2.168 (0.747–6.291) 0.015 | 2.65 (0.527–13.36) 0.237 | 1.81 (0.625–5.265) 0.273 | 2.49 (0.739–8.420) 0.141 |
| • 2 | 4.803 (1.531–15.065) 0.007 | 4.37 (0.557–34.39) 0.161 | 3.22 (1.062–9.762) 0.039 | 4.34 (0.918–20.558) 0.064 |

HR: hazard ratio; CI: confidence interval; ECOG: European Colleague Oncology Group; Non-SQ: non-squamous; irAEs: irreversible adverse events; NLR: neutrophil–lymphocyte ratio; PLR: platelet–lymphocyte ratio; LIPI: Lung Immune Prognostic Index; Pdl1: platelet-derived lymphocyte; mGPS: modified Glasgow prognostic score; N/A: not applicable.

This study obtained a dynamic mGPS status, and we further analysed the relationship between dynamic mGPS and PFS, OS, and RR, with the following results: PFS on mGPS1 (HR = 2.168, 95% CI, 0.747–6.291; *p* = 0.015) and PFS on mGPS2 (HR = 4.803; 95% CI, 1.531–15.065; *p* = 0.007); OS on mGPS1 (HR = 1.81; 95% CI, 0.625–5.265; *p* = 0.273) and OS on mGPS2 (HR = 3.22; 95% CI. 1.062–9.762; *p* = 0.039); and RR on mGPS 1 (HR = 0.789; 95% CI, 0.113–5.528; *p* = 0.812) and RR on mGPS2 (HR = 0.214; 95% CI, 0.021–2.187; *p* = 0.194). Significant associations were found between the mGPS1 and mGPS2 cut-off values and increased PFS benefit (*p* = 0.155 and *p* = 0.007, respectively). Although the mGPS cut-off value appeared to be associated with increased HRs of OS, only mGPS2 was statistically significant (*p* = 0.039). The HRs of PFS showed significant correlations with the HRs of OS, suggesting that PFS could be a potential surrogate for OS in these study designs.

**Table 3.** Univariate and multivariate analyses of clinicopathologic factors and immune-inflammation-nutritional parameters.

| Variables | OR for Response | |
|---|---|---|
| | Unadjusted OR (95% CI), p | Adjusted OR (95% CI), p |
| ECOG PS $\geq$ 2 | 0.361 (0.147–0.884) 0.026 | 0.338 (0.112–1.017) 0.054 |
| Histology-SQ | | - |
| >75 years of age | | - |
| Presence of brain metastasis | 0.977 (0.205–4.670) 0.977 | - |
| Presence of bone metastasis | 0.737 (0.298–1.822) 0.509 | |
| Presence of adrenal gland metastasis | 0.327 (0.096–1.108) 0.073 | - |
| Presence of malignant pleural metastasis | 0.844 (0.270–2.636) 0.771 | - |
| Presence of liver metastasis | 0.631 (0.222–1.793) 0.388 | - |
| irAEs | 3.007 (0.989–9.143) 0.052 | |
| NLR | 0.703 (0.556–0.888) 0.003 | 0.737 (0.513–1.059) 0.099- |
| PLR | 0.995 (0.994–1.000) 0.001 | 1.000 (0.995–1.005) 0.974- |
| LIPI status<br>● Good<br>● Intermediate<br>● Poor | N/A 0.002<br>1<br>0.138 (0.24–1.82) 0.001<br>0.196 (0.17–2.10) 0.009 | N/A, 0.015<br>1<br>0.188 (0.053–0.664) 0.009<br>0.956 (0.195–4.693) 0.956- |
| Pdl1<br>● 0<br>● 1–49<br>● >50 | | |
| CRP level | 0.961 (0.836–1.104) 0.574 | |
| mGPS<br>● 0<br>● 1<br>● 2 | N/A 0.269<br>1 (reference)<br>0.789 (0.113–5.528) 0.812<br>0.214 (0.021–2.187) 0.194 | - |

OR: odds ratio; CI: confidence interval; ECOG: European Colleague Oncology Group; SQ: squamous; irAEs: irreversible adverse events; NLR: neutrophil–lymphocyte ratio; PLR: platelet–lymphocyte ratio; LIPI: Lung Immune Prognostic Index; Pdl1: platelet-derived lymphocyte; mGPS: modified Glasgow prognostic score; N/A: not applicable.

## 4. Discussion

The aim of this study, with multiple post-treatment and immune-based prognostic scores, was to investigate the prognostic role of post-treatment 6th-week NLR, PLR, LIPI, and mGPS scores in patients with locally advanced or metastatic lung cancer. Pre-treatment NLR, PLR, LIPI, and mGPS are manifestations of baseline immune function. Their post-treatment results are theoretically modifiable factors that could be influenced by several factors, such as radiation prescriptions or therapy dosing. These findings may indicate a potential predictive marker of response. Eighty-three patients were examined, and showed that post-treatment NLR, PLR, LIPI, and mGPS were statistically significantly associated with poor prognosis in the study population. Besides determining a predictive value, our data also demonstrate an independent association with survival.

Several research articles and meta-analyses have been published about the prognostic effect of pre-treatment NLR in lung cancer, but changes in NLR status depending on treatment have not yet been determined. Our hypothesis was that the post-treatment NLR and NLR dynamics after immunotherapy would be prognostic. In our study, the patients with post-treatment NLR values (>5) up to the threshold had shorter PFS, shorter OS, and lower RR, consistent with all previous study results. In the univariate analysis, all inflammatory parameters were independent prognostic indicators, but in the multivariate analysis,

only OS was relevant to NLR. In one study, 54 patients with NSCLC were treated with anti-PD-1 treatment, NLR was assessed at baseline and 6 weeks, and low post-treatment NLR (>5) and immune-related adverse events were significantly associated with low RR and shorter PFS and OS. Liver metastasis was also an independent prognostic indicator of shorter PFS [20]. Another study on patients with NSCLC who received conventional chemotherapy and gefitinib demonstrated that an early reduction in NLR is a surrogate marker of survival [21].

The backbone treatment for patients with advanced NSCLC is platinum with cytotoxic chemotherapy [22]. We know about chemotherapy-induced neutropenia associated with increased survival in patients with advanced NSCLC [23]. Neutropenia should be a surrogate marker of chemotherapy efficacy, and deficient neutropenia in patients may indicate insufficient dosing and inadequate tumour elimination [24]. Neutrophils can also be manipulated to develop different functional polarization and phenotypic states to induce anti- or pro-tumour effects in the tumour microenvironment [25]. Finally, the patterns of NLR change after the 6th week of treatment as a prognostic factor for PFS and OS were consistent with the immunotherapy treatment regimens.

High platelet levels play an active role in inflammation, tissue regeneration, or acceleration of tumour progression [26]. By contrast, lymphocytes release some types of cytokines that activate anti-tumour immunity [27]. Recently, elevated PLR was shown to be closely related to poor prognosis in various solid tumours [28]. Our study shows that the PLR level elevation in the 6th week of ICI treatment was significantly associated with the initial response, PFS, and OS of ICI treatment. We should speculate that differences between studies in terms of cancer type, demographic specialities, treatment modalities, sample size, and the threshold PLR value used for bisection might have been responsible. In patients with NSCLC treated predominately with nivolumab or pembrolizumab, higher PLR correlated with worse OS [29]. A meta-analysis of 12 studies reported that pre-treatment PLR could be a routine potential prognostic factor and have a predictive role concerning the survival of patients with cancer treated with immunotherapy [30]. Another two studies related to NSCLC found no significant difference in survival between patients with NSCLC with high and low baseline PLR levels [31,32].

In 2018, Mezquita et al. [33] developed a new potential blood-based biomarker, LIPI, which stratified baseline dNLR and LDH in patients with NSCLC under anti-PDL1 treatment according to survival outcomes. Previous studies have indicated that LIPI could predict clinical outcomes across many tumour types, such as renal cell carcinoma, melanoma, small-cell lung cancer, and especially NSCLC. However, the prognostic value of LIPI remains a divisive issue. Mostly, the combination of baseline dNLR and LDH correlated with resistance to ICI therapy in patients with advanced NSCLC. We also explored the predictive value of LIPI in these contexts. The present study shows that the intermediate LIPI group had significantly different RRs compared with the poor group during the 6th week of ICI treatment. In addition, the poor LIPI group had worse OS and PFS after ICI treatment compared with the good group.

The LDH level is a known prognostic inflammatory marker in patients with cancer and has been widely studied in patients with lung cancer treated with chemotherapy or patients with EGFR-mutant NSCLC. LDH was associated with DCR, PFS, and OS during the first month of erlotinib treatment [34]. Neutrophils are a crucial component of inflammation, playing an essential role in initiating tumorigenesis by damaging specific tissues. In cancer, neutrophils can promote or prevent tumour progression. Both increased and decreased neutrophil counts have been associated with tumour initiation [35]. Another mechanism is the neutrophil pro-inflammatory status, which induces uncontrolled granulopoiesis, releasing immature or poorly differentiated neutrophils, and has been associated with tumour progression [36].

mGPS is a composite biomarker that reflects both host-related systemic inflammatory response and nutritional status. Some studies have shown that baseline mGPS is an independent prognostic factor for PFS and OS in patients with advanced NSCLC treated

with anti-PD1 treatment [37]. The exact mechanisms of inflammation related to prognosis remain unclear [3]. One of the suggested pathways is linked with GPS. Increasing data have shown that the presence of patient-related factors, especially nutritional and functional statuses, is associated with poorer outcomes in addition to the tumour stage. mGPS has been used as a biomarker to reflect the degree of cancer-associated inflammation and malnutrition. mGPS is a kind of systemic inflammatory response (SIR)-based scoring system that combines the indicators of decreased plasma albumin and elevated CRP [18]. The mGPS has been evaluated as a prognostic parameter in accordance with findings in various malignancies [38]. Serum CRP levels, which are identified by the activation of proinflammatory cytokines, might lead to tumour invasion, progression, and the formation of metastases [39]. Most studies have shown possible relationships between chronic, systemic inflammatory response, compromised cellular immune response [40], and tumour cachexia [41] caused by low serum albumin levels. In our study, the 6th-week mGPS showed no independent association with OS and PFS. In the literature, only one study investigated the association of 6th-week mGPS with PFS and OS in patients with metastatic renal cell carcinoma who were treated with sunitinib [42]. We need studies involving a larger number of patients to clarify the role of 6th-week mGPS in NSCLC patients.

Another aspect of our study is that the presenting metastasis site was associated with the outcome of the anti-PD-1 antibody treatment. Liver metastasis (HR = 3.093; 95% CI, 1.017–9.405; *p* = 0.047) was an independent prognostic indicator of shorter PFS. As we know, the liver has an immunological organ interplay between immune tolerance and immune activation, which provides for the development of novel therapeutic strategies for cancer [43]. Kupfer cells, liver sinusoidal endothelial cells, and dendritic cells play important roles in reducing the immune response and maintenance of immune-suppressive status [44]. The poor response to and shorter PFS with anti-PD-1 antibody treatment in patients with liver metastases could be explained by the maintenance of immunotolerance.

Our study has several limitations. Its retrospective design and relatively small sample size limited the significance of the subgroup analysis. PD-L1 analyses were not available in most patients.

In daily clinical practice, oncologists expect to quickly determine the treatment response because lung cancer usually develops extremely progressively in this state. The measurement of serum inflammatory parameters is non-invasive and inexpensive in the assessment of the efficacy of immunotherapy treatment in patients with lung cancer. Quickly increasing inflammatory markers may be related to primary refractory disease, indicating a poorer prognosis. In patients with negative prognostic results in the biomarker analysis at 6 weeks of treatment, imaging will need to be re-evaluated earlier to catch a progression of disease as soon as possible. If the 6th-week post-treatment NLR, PLR, LIPI, and mGPS ratio tend to decrease, this may reassure physicians that they are on the right track to directing treatment response and better survival.

**Author Contributions:** Conceptualization, P.O. and O.D.; methodology, P.O.; software, formal analysis, P.O.; investigation, P.O. and O.D.; data curation, P.O. and O.D.; writing—original draft preparation, P.O.; writing—review and editing, P.O. and O.D.; visualization, P.O.; supervision, P.O. and O.D.; project administration, P.O. All authors have read and agreed to the published version of the manuscript.

**Funding:** This research received no external funding.

**Institutional Review Board Statement:** This study was conducted according to the guidelines of the Declaration of Helsinki and approved by the Local Ethical Committee of Burhan. Nalbantoğlu Research Hospital approved the study in April 2021 with decision number E-21/21.

**Informed Consent Statement:** All participants provided written informed consent for the storage of medical information in the hospital database and the use of this information for research purposes.

**Data Availability Statement:** Data cannot be shared publicly because the data are owned and saved by Burhan Nalbantoglu Research Hospital and Near East University Hospital. Datasets are available on request from the corresponding author.

**Conflicts of Interest:** The authors declare that they have no conflict of interest.

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
