# Peer review of "Sixth-Week Immune-Nutritional-Inflammatory Biomarkers: Can They Predict Clinical Outcomes in Patients with Advanced Non-Small Cell Lung Cancer Treated with Immune Checkpoint Inhibitors?"

_curroncol, doi:10.3390/curroncol30120769_

Round 1

Reviewer 1 Report

Comments and Suggestions for Authors

In this manuscript, authors study how immune-nutritional-inflammatory biomarkers can predict clinical outcomes in patients with advanced NSCLC treated with immune checkpoint inhibitors. Although the study is interesting, the results presented need to be accompanied by histological data to complete their study. Authors should also improve the materials and methods section with more details. Introduction should also be improved with more information. 

Author Response

Material, method and introduction has improved with more information. All corrections are highlighted in yellow.

Reviewer 2 Report

Comments and Suggestions for Authors

Some suggestions for the Authors:

> line 34-35: they do not mention radiotherapy, which is crucial in the treatment of lung cancer;

> line 42-45 the phrasing is not correct; the sentence does not come to a logical conclusion; please, re-phrase;

> the authors include patients with EGFr mutation and ALK re-arrangement; why did they choose to treat these patients with ICI? It is widely known of the poor outcome of ICI in oncogene-addicted patients.

> How did they choose the threshold for biomarker values? Did they perform a ROC analysis?

> Authors chose 6 week as a timepoint; why did they choose such timepoint? Is there a biological rationale? 

> An analysis at 6 weeks after the beginning of ICIs does not have a value in choosing the therapy? Did they perform a similar analysis at baseline? How did the values change in responders vs non responders at 6 weeks?

> An analysis at 6 weeks after the beginning might influence the timing of imaging re-evaluation to catch a progression of disease as soon as possible.  They should then suggest that patients experiencing negative prognostic results at the biomarker analysis at 6 weeks should be evaluated earlier than the other ones. They should include these statements in the conclusions. 

Comments on the Quality of English Language

The manuscript is peppered with grammar mistakes, incorrect word choices, and wordy phrasing. It should be revised by an English native speaker. 

Author Response

> line 34-35: they do not mention radiotherapy, which is crucial in the treatment of lung cancer;

Answer: corrected

> line 42-45 the phrasing is not correct; the sentence does not come to a logical conclusion; please, re-phrase;

Answer: Revised by an English native speaker(scribendi proofreading services)

> the authors include patients with EGFr mutation and ALK re-arrangement; why did they choose to treat these patients with ICI? It is widely known of the poor outcome of ICI in oncogene-addicted patients.

Answer:corrected

> How did they choose the threshold for biomarker values? Did they perform a ROC analysis?

Answer: We did not perform ROC analysis. Our threshold for biomarker values were determined based on previus similar studies.

> Authors chose 6 week as a timepoint; why did they choose such timepoint? Is there a biological rationale? 

Answer: ‘Little information is available (in 2019, Kasahara et al. [13]; in 2021, Küçükarda et al. [14]) regarding the association of the post-treatment inflammation parameter with response to anti-PD1 treatment in patients with advanced NSCLC.’

> An analysis at 6 weeks after the beginning of ICIs does not have a value in choosing the therapy? Did they perform a similar analysis at baseline? How did the values change in responders vs non responders at 6 weeks?

Answer: ‘In this study, we aimed to determine the prognostic use of posttreatment NLR, PLR, LIPI, and mGPS
obtained at the 6th week.Many studies have been conducted on the prognostic importance of pretreatment inflammatory parameters.’ We did not compared  both of them.

> An analysis at 6 weeks after the beginning might influence the timing of imaging re-evaluation to catch a progression of disease as soon as possible.  They should then suggest that patients experiencing negative prognostic results at the biomarker analysis at 6 weeks should be evaluated earlier than the other ones. They should include these statements in the conclusions. 

Answer:

In daily clinical practice, oncologists expect to quickly determine the treatment response because lung cancer usually develops extremely progressively in this state. The measurement of serum inflammatory parameters is non-invasive and inexpensive in the assessment of the efficacy of immunotherapy treatment in patients with lung cancer. Quickly increasing inflammatory markers may be related to primary refractory disease, indicating a poorer prognosis. In patients with negative prognostic results in the biomarker analysis at 6 weeks of treatment, imaging will need to be re-evaluated earlier to catch a progression of disease as soon as possible. If the 6th-week post-treatment NLR, PLR, LIPI, and mGPS ratio tend to decrease, this may reassure physicians that they are on the right track to directing treatment response and better survival.

The manuscript is peppered with grammar mistakes, incorrect word choices, and wordy phrasing. It should be revised by an English native speaker. 

Revised by an English native speaker. (scribendi proofreading services)

All corrections are highlighted in yellow 

Reviewer 3 Report

Comments and Suggestions for Authors

 Polat Olgun and Omer Diker are describing Sixth-week immune-nutritional-inflammatory biomarkers: which were checked as a predict clinical outcomes in patients with advanced non-small-cell lung cancer treated with immune checkpoint inhibitors.

Interesting idea and well-organized work. But I have few comments and more information need to be understood!-

  1-Please add more information in the introduction regarding lung cancer.

In the following article-"Szeto, C.H.; Shalata, W.; Yakobson, A.; Agbarya, A. Neoadjuvant and Adjuvant Immunotherapy in Early-Stage Non-Small-Cell Lung Cancer, Past, Present, and Future. J. Clin. Med. 2021, 10, 5614. https://doi.org/10.3390/jcm10235614".

It was mentioned "Non-small-cell lung cancer (NSCLC) is the most common type of lung cancer, accounting for 85% of all lung cancer diagnoses. Histologically, NSCLC is divided into three main types: adenocarcinoma, squamous cell carcinoma, and large cell carcinoma. NSCLC is often insidious and undiagnosed until advanced-stage disease is present. Approximately 25% of patients with NSCLC have localized disease at the time of diagnosis. Lobectomy followed by systemic adjuvant therapy is considered as standard treatment for patients with resectable NSCLC. Despite the standard treatment, 50% of patients with stage II, and 60% of patients with stage IIIA disease die within five years. Therefore, researchers have been exploring novel treatment approaches to reduce the risk of recurrence and improve survival of resectable NSCLC." you can use information from there

and also include the following two articles:

Wang, L., Hu, Y., Wang, S., Shen, J., & Wang, X. Biomarkers of immunotherapy in non-small cell lung cancer. Oncol Lett. 2020; 20(5); 139. https://doi.org/10.3892/ol.2020.11999

Gridelli C, Ardizzoni A, Barberis M, et al. Predictive biomarkers of immunotherapy for non-small cell lung cancer: results from an Experts Panel Meeting of the Italian Association of Thoracic Oncology. Transl Lung Cancer Res. 2017;6(3):373-386. doi:10.21037/tlcr.2017.05.09

 2-Regarding to lines 51-52 " patients with solid tumors" please write/add which were?

The last updated article regarding the immune nutritional-inflammatory biomarkers was " Yakobson, A.; Abu Jama, A.; Abu Saleh, O.; Michlin, R.; Shalata, W. PD-1 Inhibitors in Elderly and Immunocompromised Patients with Advanced or Metastatic Cutaneous Squamous Cell Carcinoma. Cancers 2023, 15, 4041. https://doi.org/10.3390/cancers15164041".

There it was mentioned, "In solid tumors, the inflammatory response can be characterized by various parameters in the peripheral blood, including baseline leukocytes and their subtypes, C-reactive protein, plasma fibrinogen, the neutrophil-to-lymphocyte ratio, albumin, alkaline phosphatase, and the lymphocyte-to-monocyte ratio. These parameters have been widely discussed as prognostic indicators in many solid tumors, especially in skin cancers (cutaneous melanoma)" but in Cutaneous Squamous Cell Carcinoma it showed no significant differences in the PFS results.

Please add these information to the main text.

 3- Table 1 please add the parameters on the upper right of the table –number (%).

 4- Table 1- What was the meaning of "Immunotherapy line step" did you mean cycle? If yes pleas right cycle and not step.

5- is giving immunotherapy changed the results? Did you give alone (monotherapy) single agent as 1st line of therapy of metastatic disease?

Monotherapy/combo-immunotherapy was different from chemo+IO?

 6- in "Study population" you mentioned only anti-PD-1 antibody (nivolumab or pembrolizumab) what about antiCLA4- ipilimumab? Please add it. (I saw that it was mentioned in Table 1)

 7- also "Data Collectionltion" add the informations regarding ipilimumab. (I saw that it was mentioned in Table 1)

 8- how could it be that "PDL-1" status is unknown? If it is unknown how the treatment was chosen for 77.1 % of patients? It is kind of weird!

Comments on the Quality of English Language

need to be better written.

Author Response

1-Please add more information in the introduction regarding lung cancer.

In the following article-"Szeto, C.H.; Shalata, W.; Yakobson, A.; Agbarya, A. Neoadjuvant and Adjuvant Immunotherapy in Early-Stage Non-Small-Cell Lung Cancer, Past, Present, and Future. J. Clin. Med. 2021, 10, 5614. https://doi.org/10.3390/jcm10235614".

It was mentioned "Non-small-cell lung cancer (NSCLC) is the most common type of lung cancer, accounting for 85% of all lung cancer diagnoses. Histologically, NSCLC is divided into three main types: adenocarcinoma, squamous cell carcinoma, and large cell carcinoma. NSCLC is often insidious and undiagnosed until advanced-stage disease is present. Approximately 25% of patients with NSCLC have localized disease at the time of diagnosis. Lobectomy followed by systemic adjuvant therapy is considered as standard treatment for patients with resectable NSCLC. Despite the standard treatment, 50% of patients with stage II, and 60% of patients with stage IIIA disease die within five years. Therefore, researchers have been exploring novel treatment approaches to reduce the risk of recurrence and improve survival of resectable NSCLC." you can use information from there

and also include the following two articles:

Wang, L., Hu, Y., Wang, S., Shen, J., & Wang, X. Biomarkers of immunotherapy in non-small cell lung cancer. Oncol Lett. 2020; 20(5); 139. https://doi.org/10.3892/ol.2020.11999

Gridelli C, Ardizzoni A, Barberis M, et al. Predictive biomarkers of immunotherapy for non-small cell lung cancer: results from an Experts Panel Meeting of the Italian Association of Thoracic Oncology. Transl Lung Cancer Res. 2017;6(3):373-386. doi:10.21037/tlcr.2017.05.09

Answer:  Lung cancer is the most prevalent, life-threatening malignancy and cancer-related death worldwide [1]. The primary traditional treatments for patients with lung cancer are surgery ,radiotherapy and chemotherapy. Approximately %75 of cases are diagnosed at advanced stages, and the benefits achieved from chemotherapy and the prognosis of non-small-cell lung cancer (NSCLC) remain poor in advanced stages.  NSCLC is classified into three main histological types: adenocarcinoma, squamous cell carcinoma, and large cell carcinoma.[1.1]

 2-Regarding to lines 51-52 " patients with solid tumors" please write/add which were?

The last updated article regarding the immune nutritional-inflammatory biomarkers was " Yakobson, A.; Abu Jama, A.; Abu Saleh, O.; Michlin, R.; Shalata, W. PD-1 Inhibitors in Elderly and Immunocompromised Patients with Advanced or Metastatic Cutaneous Squamous Cell Carcinoma. Cancers 2023, 15, 4041. https://doi.org/10.3390/cancers15164041".

There it was mentioned, "In solid tumors, the inflammatory response can be characterized by various parameters in the peripheral blood, including baseline leukocytes and their subtypes, C-reactive protein, plasma fibrinogen, the neutrophil-to-lymphocyte ratio, albumin, alkaline phosphatase, and the lymphocyte-to-monocyte ratio. These parameters have been widely discussed as prognostic indicators in many solid tumors, especially in skin cancers (cutaneous melanoma)" but in Cutaneous Squamous Cell Carcinoma it showed no significant differences in the PFS results.

Please add these information to the main text.

Answer: ‘Tumor-associated inflammation in patients with cancer is believed to influence the host immune response and resistance,  growth, and migration of tumors via certain inflammatory factors [4]. Due to the interplay between systemic inflammation, the immune system, and immunotherapy, pretreatment neutrophil-to-lymphocyte ratio (NLR), platelet-to-lymphocyte ratio (PLR), Lung Immune Prognostic Index (LIPI), and modified Glasgow Prognostic Scores (mGPS) have been indicated to predict the therapeutic effect or outcomes related to poor survival in patients with various cancers (gastrointestinal carcinomas, renal cancer,breast cancer,ovarian cancer, prostate cancer etc.) [5-7].’

 3- Table 1 please add the parameters on the upper right of the table –number (%).

corrected

 4- Table 1- What was the meaning of "Immunotherapy line step" did you mean cycle? If yes pleas right cycle and not step.

corrected

5- is giving immunotherapy changed the results? Did you give alone (monotherapy) single agent as 1st line of therapy of metastatic disease?

Monotherapy/combo-immunotherapy was different from chemo+IO?

It was stated that table 1 combo immunotherapy was given to only 1 patient. And added ‘In the first line, patients with pDL1 ³50% received mono immunotherapy, while patients with pDL1 <50% received maintenance immunotherapy after 4 cycles of chemotherapy and immunotherapy.’

I had also mentioned anti-neoplastic therapies on data collection.                 

 6- in "Study population" you mentioned only anti-PD-1 antibody (nivolumab or pembrolizumab) what about antiCLA4- ipilimumab? Please add it. (I saw that it was mentioned in Table 1)

Corrected

We conducted a retrospective analysis of consecutive patients with NSCLC who had undergone anti-PD-1 antibody (nivolumab or pembrolizumab) and anti CTLA4 (ipilimumab) treatment at Burhan Nalbantoglu Research hospital and Near East University hospital between March 2017 and March 2021.

 7- also "Data Collectionltion" add the informations regarding ipilimumab. (I saw that it was mentioned in Table 1)

Corrected

The inclusion criteria were as follows: (a) pathologically confirmed NSCLC; (b) initial stage IIIB or IV, recurrence after curative surgery or maintenance after chemoradiotherapy; and (c) administration of nivolumab at 3 mg/kg every 2 weeks ,pembrolizumab at  200-mg flat dose every 3 weeks or 4 cycles nivolumab at 3 mg/kg and ipilimumab at 1 mg/kg every 3 weeks then maintenance nivolumab at 3 mg/kg every 2 weeks  as palliative therapy. All patients had anti-PD-1 therapy and some received additional anti-neoplastic therapies. Baseline patient and tumor characteristics was showed in Table 1.

 8- how could it be that "PDL-1" status is unknown? If it is unknown how the treatment was chosen for 77.1 % of patients? It is kind of weird!

Adding ‘Our study has several limitations. The retrospective design and relatively small sample size limited the significance of the subgroup analysis. PD-L1 analyses were not available in most of the patients.’

All corrections highlighted in yellow

Round 2

Reviewer 1 Report

Comments and Suggestions for Authors

Authors did not address all my concerns. 

Author Response

'the results presented need to be accompanied by histological data to complete their study.'

Answer:

There are few studies on predictive and prognostic parameters of post treatment immune inflammation. Corrections were made to the introduction and discussion sections as stated below.

1-) Little information is available (in 2019, Kasahara et al. [13]; in 2021, Küçükarda et al. [14]) regarding the association of the post-treatment inflammation parameter with response to anti-PD1 treatment in patients with advanced NSCLC.

2-)In our study, 6th week mGPS was shown no independent association with OS and PFS. In literature, only one study that investigate association of 6th week mGPS associated with PFS and OS in patients with metastatic renal cell carcinoma that were treated with sunitinib [43]. We need studies involving a larger number of patients to clarify the role of 6th week mGPS in NSCLC patients.

 'Authors should also improve the materials and methods section with more details. Introduction should also be improved with more information. '

Answer;

Introduction ,material, method, results ,discussion and references has improved with more information. All corrections are highlighted in yellow.

Reviewer 2 Report

Comments and Suggestions for Authors

I find that the Authors have correctly replied to my previous comments.

Author Response

Some corrections have been made with other reviewer suggestions.The last
version of manuscript is attached.
Sincerely regards;

Reviewer 3 Report

Comments and Suggestions for Authors

Interesting article, showing how follow up with patients can help, it can give us a backbone as oncologists, to keep following our patients in such a situation. Also it gives us a new ideas for follow up, and for choosing the treatment regarding to blood tests 

Interesting article and results

Author Response

The final version of the manuscript is attached. Thank you for the suggestions.
Sincerely regards;

Round 3

Reviewer 1 Report

Comments and Suggestions for Authors

Authors have addresed all my concerns